# Toward Using Wearables to Remotely Monitor Cognitive Frailty in Community-Living Older Adults: An Observational Study

**DOI:** 10.3390/s20082218

**Published:** 2020-04-14

**Authors:** Javad Razjouyan, Bijan Najafi, Molly Horstman, Amir Sharafkhaneh, Mona Amirmazaheri, He Zhou, Mark E. Kunik, Aanand Naik

**Affiliations:** 1VA HSR&D Center for Innovations in Quality, Effectiveness and Safety, Michael E. DeBakey VA Medical Center, Houston, TX 77004, USA; Javad.Razjouyan@bcm.edu (J.R.); mhorstma@bcm.edu (M.H.); mkunik@bcm.edu (M.E.K.);; 2Department of Medicine, Baylor College of Medicine, Houston, TX 77004, USA; amirs@bcm.edu; 3Interdisciplinary Consortium on Advanced Motion Performance (iCAMP), Division of Vascular Surgery and Endovascular Therapy, Michael E. DeBakey Department of Surgery, Baylor College of Medicine, Houston, TX 77004, USA; monaliz82@gmail.com (M.A.); He.Zhou2@bcm.edu (H.Z.); 4VA South Central Mental Illness Research, Education and Clinical Center, Houston, TX 77004, USA

**Keywords:** cognitive frailty, motoric cognitive risk syndrome, wearable, cognitive impairment, telehealth, remote patient monitoring, digital health

## Abstract

Physical frailty together with cognitive impairment (Cog), known as cognitive frailty, is emerging as a strong and independent predictor of cognitive decline over time. We examined whether remote physical activity (PA) monitoring could be used to identify those with cognitive frailty. A validated algorithm was used to quantify PA behaviors, PA patterns, and nocturnal sleep using accelerometer data collected by a chest-worn sensor for 48-h. Participants (*N* = 163, 75 ± 10 years, 79% female) were classified into four groups based on presence or absence of physical frailty and Cog: PR-Cog-, PR+Cog-, PR-Cog+, and PR+Cog+. Presence of physical frailty (PR-) was defined as underperformance in any of the five frailty phenotype criteria based on Fried criteria. Presence of Cog (Cog-) was defined as a Mini-Mental State Examination (MMSE) score of less than 27. A decision tree classifier was used to identify the PR-Cog- individuals. In a univariate model, sleep (time-in-bed, total sleep time, percentage of sleeping on prone, supine, or sides), PA behavior (sedentary and light activities), and PA pattern (percentage of walk and step counts) were significant metrics for identifying PR-Cog- (*p* < 0.050). The decision tree classifier reached an area under the curve of 0.75 to identify PR-Cog-. Results support remote patient monitoring using wearables to determine cognitive frailty.

## 1. Introduction

Cognitive impairment and physical frailty are known as two independent geriatric risk factors for adverse clinical outcomes including hospitalization, disability, loss of independence, dementia, morbidity, and mortality [1,2], as well as future cognitive decline [3,4,5]. There is growing awareness and evidence that physically frail older adults are prone to cognitive impairment and vice versa. Studies have shown that the coexistence of these two geriatric conditions (cognitive frailty) creates a vicious cycle of declines in older adults’ quality of life [6,7,8]. To emphasize the importance of screening for cognitive frailty, the International Consensus Group on “Cognitive Frailty”, organized by The International Academy on Nutrition And Aging (I.A.N.A) and the International Association of Gerontology and Geriatrics (I.A.G.G), defined four aging trajectories: (1) older adults without physical frailty or cognitive dysfunction; (2) older adults with physical frailty but with normal cognitive functioning; (3) older adults with no physical frailty but exhibiting cognitive impairment; and (4) older adults with both physical frailty and cognitive impairment (cognitive frailty) [9]. To our knowledge, there is currently no remote patient monitoring system that objectively monitors cognitive frailty.

Common approaches for assessing cognitive frailty are based on assessing cognition [10,11] and physical frailty [12,13] under standardized conditions. These assessments require considerable resources, are limited to one-off periodic assessments, may suffer bias associated with obtrusive observation (e.g., “white coat” effect), and are often subjective or semi-objective and prone to practice effects, such as patient educational level and/or native speaking language [14,15]. Most importantly, current approaches are cumbersome, performed in clinical settings, and use tools with limited capacity to remotely monitor and track changes in cognitive frailty over time, which may not be practical for older adults who are living in remote areas [16,17].

To overcome these limitations for assessing physical frailty, wearable sensors have been proposed as a more practical tool to address issues such as feasibility, practicality, ease of use, accessibility, reproducibility, and reliability, without hindering daily activity in outpatient or inpatient settings [18,19,20]. Activity-based sensor data have shown promise for measuring physical frailty using activity-related metrics such as steps [15,18,19,20] or sleep quantities such as total sleep time [17,21,22]. Previous studies in community dwelling settings by Schwenk et al. have shown that multiple sensor-based physical activity monitors, which measure posture (walking, standing, sitting, lying, and postural transition from sit-to-stand and stand-to-sit) and supervised gait parameters (stride length, gait speed, gait velocity, and cadence), are capable of discriminating between non-frail, pre-frail, and frail patients [20]. Other studies have shown that sensor-derived activity levels (sedentary behaviors, light and moderate-to-vigorous activity) are highly correlated with frailty status [23] and are capable of discriminating between different frailty states [24,25,26,27]. They found that an increase in sedentary behavior and a decrease in high-intensity activity, such as moderate-to-vigorous activity, represent a strong predictor of frailty progression. Interestingly, Theou et al. showed that a single parameter, the number of steps (derived from a wearable sensor), is significantly correlated with the progression of frailty [26].

Several studies have used activity-based sensor data to assess the physical frailty based on sleep quantities [17,21,22]. These studies showed that the robust patient has fewer sleep disruptions. Ensrud et al. 2012 showed that robust patients with sleep disruption have higher odds of becoming frail after 3 to 4 years follow-up [21]. A recent cross-sectional study by Razjouyan et al. 2018 showed that frail patients have impaired sleep quantities compared to robust and pre-frail patients [17].

Despite promising progress in remote monitoring of physical frailty using wearables, few studies have demonstrated the ability of wearables to remotely monitor cognitive decline. Several studies suggested that remote monitoring of activities of daily living (ADL) or gait may be used as surrogate markers for cognitive decline [20,28,29]. Some studies have used wearable sensor data from in-home monitors to assess cognitive impairment based on sleep quantities [30,31,32,33]. A systematic review showed that older adults who have a sleep duration of 7–8 h per day have a lower risk of cognitive disorders [33]. Another prospective cohort study on women showed that participants with disrupted sleep quantities (i.e., lower sleep efficiency, longer sleep latency, variability in sleep time) are at higher risk of developing cognitive impairment [32]. Several other proof-of-concept studies found that patients with cognitive impairment have disrupted sleep–wake patterns [32] or circadian rhythm disorders [31].

While prior studies showed the ability of wearable sensors to identify age-related “motoric cognitive risk syndrome” [34] (decline in both cognitive and motor performance), they did not explore whether wearable sensors can distinguish between the four frailty groups, especially the cognitive frailty group. In this study, we aim to evaluate the ability of sensor-derived parameters to distinguish older adults at risk for cognitive frailty syndrome from those who are not at risk. We hypothesize that indicators of mobility performance (e.g., daily step counts, activity behavior, and commutative postures such as sitting, standing, and lying) and sleep quantity (e.g., total sleep time, sleep onset latency, etc.), measured remotely using a pendant sensor, could identify those who suffer from cognitive frailty syndrome.

## 2. Methods

### 2.1. Participants

We recruited community-living older adults ≥ 60 years of age without severe gait or balance disorders (i.e., able to independently walk a distance of at least 15 feet (~4.5 m) with or without walking assistance). Exclusion criteria were severe cognitive impairment (Mini-Mental State Examination (MMSE) score ≤ 16) [35], severe visual or hearing problems, unstable medication use in over the previous last 6 weeks, and acute conditions (e.g., recent surgery, active foot ulcer, recent stroke, etc.). Participants who met the eligibility criteria signed written consent forms. Demographic and clinical traits, such as age, body mass index (BMI), and clinical characteristics such as number of prescribed medications, depression as assessed by the Center for Epidemiologic Studies Depression Scale (CSE-D) [36], and concern for fall (The Falls Efficacy Scale International, [FES-I]) [37] were collected. This study was approved by the local institutional review boards at the Baylor College of Medicine Institutional Review Board (IRB protocol: H-38994). All the clinical assessments were performed at the time of clinical visits, and the sensors were provided at the same time.

### 2.2. Physical Frailty and Cognitive Impairment Assessment

The frailty phenotypes developed by Fried et al. [12] served as the standard criterion for assessing the physical frailty status of participants, consisting of five core clinical frailty criteria: shrinking (self-reported unintentional weight loss ≥ 10 lb (approximately ~4.5 kg) in the previous year), exhaustion (using two questions from the Center for Epidemiologic Studies Depression Scale (CES-D)), inactivity (short version of Minnesota Leisure Time Activity questionnaire), slowness (walking time in seconds over 15 feet (approximately ~4.5 m) at usual pace), and weakness (hand grip test) [12]. Individuals meeting the threshold of impairment in any of the five criteria were classified as “not physically robust (PR-)”.

Similarly, we defined cognitive impairment risk (Cog–) as a MMSE score of less than 27 as recommended by Luck et al. (2015) and Damian et al. (2011) for distinguishing older adults who were likely to develop cognitive impairment [38,39]. A previous study [38] showed that the cutoff-point of 27 and lower would produce the highest sensitivity and specificity (measured by Youden’s J statistic [40] = sensitivity+specificity-1) compared to clinical gold standard clinical neurological tests. In another study [39], the authors used an MMSE score of ≥ 28 to include participants with intact global cognitive function. They found that this cut-off best distinguishes between cognitive normalcy and cognitive impairment compared to commonly used lower threshold scores (i.e., 25) [39].

Our four older adult cohorts were defined (Figure 1A) as follows: (1) no signs of physical frailty or cognitive impairment risk (physically robust, PR+ and low cognitive risk, Cog+), (2) no signs of physical frailty, but with cognitive impairment risk (PR+Cog-), (3) signs of physical frailty but low cognitive risk (PR-Cog+), and (4) signs of physical frailty and cognitive impairment risk (cognitively frail, PR-Cog-). The clinical assessment was performed by trained research assistants or the clinicians at the time of the visits.

### 2.3. Sensor-Derived Parameters and Non-Wear Time

We used a pendant sensor (PAMSys^TM^, BioSensics LLC, Newton, MA, USA, Figure 1B) to monitor daily physical activities and sleep. Participants were instructed to keep the sensor on for 48 h and then return it to the center through a pre-paid envelope or collection by study coordinators at the next clinical visit. The PAMSys sensor has three-dimensional accelerations (sampling frequency= 50 Hz) and the company provides software to download, calibrate, and normalize data (PAMware, Newton, MA, USA). We used validated algorithms to extract physical activities and sleep parameters, including cumulative postures (e.g., lying, sitting, standing, and walking), locomotion (e.g., step counts), activity behavior (e.g., daily percentage of sedentary behavior, light activity, and moderate to vigorous activity), and nocturnal sleep parameters (e.g., time in bed, total duration of sleep, etc.) [41,42,43,44,45]. A valid (i.e., >8 h) “day with wear-time” annotation was used to report physical activity parameters. The participants’ averages over 2 days were used to define their reported level for each variable, as this interval provides the highest inter-class correlation (ICC) [46,47,48,49,50,51,52].

We used the algorithm suggested by Razjouyan et al. [45] to extract the nocturnal sleep parameters, which is a validated algorithm for a chest-worn sensor and has been shown to have an accuracy of 85.8% to distinguish between awake and sleep epochs with 85.8% accuracy compared to the gold standard of polysomnography. The nocturnal sleep parameters extracted for the purpose of our study were: (1) Time in bed (TiB), hours: total duration of participant’s time in bed; (2) Total sleep time (TST), hours: total duration of actual time spent asleep; (3) Sleep onset latency (SOL), min: time to fall asleep from the beginning of TiB; (4) Wake after sleep onset (WASO), min: duration of wake after sleep onset till sleep offset; (5) Sleep efficiency (SE), percent %: percentage of TST/(TST + WASO); (6) Supine position, percent %: duration of supine position during TiB; (7) Prone position, percent %: duration of prone position during TiB; (8) Side-lying position, percent %: duration of side lying (left or right) during TiB; and (9) TST ≥ 6 h: the number of participants who slept more than 6 h.

Sleep parameters were averaged between two consecutive nights. The daytime physical activity parameters extracted for the purpose of our study were: (1) Cumulative postures, percent %: duration of each posture (lying, sitting, standing, walking); (2) Total steps: total number of steps per day, (3) Sedentary duration, such as sitting or lying with metabolic equivalent (MET) < 1.5; (4) Light activity, such as domestic chores with 1.5 ≤ MET < 3.0; and (5) Moderate-to-vigorous (MtV) activity, such as walking with MET ≥ 3.0. [46,53]. All the sensor-derived parameters were extracted from two continuous days (48 h) of recording. Prior studies have shown that two days of activity monitoring are sufficient to determine the frailty stages [44] and that this period yields an optimum adherence to continuously wear the sensor [54].

### 2.4. Statistics

We used Fisher’s exact test to evaluate the differences between categorical variables (demographic or clinical characteristics). The correlation between sensor-derived parameters and MMSE was calculated using Spearman correlation coefficients. We used ANCOVA with Tukey LSD post hoc test performed on SPSS (IBM, V24.0.0) to test the significance level between the three groups of PR+Cog+, PR-Cog+, and PR-Cog-. We reported *p*-values and Cohens’ *d* effect size (ES) for normally distributed parameters and Cramer’s *V* for parameters that violate the normality assumption [55].

To identify the independent sensor-derived parameters that classify PR-Cog-, we utilized an embedded feature selection approach to the whole dataset [56,57]. The independent predictors were fed to a decision tree classifier [58] to identify PR-Cog- from the rest (one-vs-rest [59]). Decision trees have gained significant attention in medical decision-making because: (1) they provide high classification accuracy and (2) they provide simple representation of gathered knowledge [60]. We used MATLAB software (MathWorks, Natick, MA, USA) to develop the model with default settings. The performance of this model is reported using sensitivity, specificity, accuracy, and area under the curve (AUC) based on K-fold cross validation (K = 10) [61]. In this approach, the dataset was sliced into 10 splits; 9 splits were used for training, and the remaining split was used for testing. The process was repeated 10 times [61]. Classifier performance was reported by calculating a 2 × 2 confusion matrix that has 4 components: true positive (TP), false positive (FP), false negative (FP), and true negative (TN). We also reported the sensitivity (=TP/(TP+FN)), specificity (=TN/(TN+FP)), accuracy (=(TP+FN)/(TP+TN+FP+FN)) and area under the curve (AUC) [62].

## 3. Results

### 3.1. Participants

A total of 163 older adults who satisfied inclusion and exclusion criteria were recruited. Based on physical frailty and cognitive function assessments, we identified four groups of participants as follows: PR+Cog- (*n* = 41), PR+Cog- (*n* = 4), PR-Cog+ (*n* = 89), and PR-Cog- (*n* = 29) (Figure 1A). Only four participants were classified as PR+Cog- and thus were excluded from group comparisons due to low sample size. Among the 159 participants (75 ± 10 years and 79% female) who remained in the study, 41 (26%) were PR+Cog+, 89 (60%) were PR–Cog+, and 29 (14%) were PR-Cog- (Table 1). As expected, the cognitive frailty group had the lowest MMSE score (24.5 ± 2.6) compared to the PR+Cog+ (29.5 ± 0.7, *p* < 0.001, ES = 2.65) and PR–Cog+ (29.2 ± 0.8, *p* < 0.001, ES = 2.49) groups. Interestingly, age was not a predictor of cognitive frailty. BMI enabled for distinguishing between the cognitive frailty and PR+Cog+ (*p* < 0.001) groups, but not from PR–Cog+ group (*p* = 0.170).

The CES-D value, the indicator of depression in the PR+Cog+ group (6.5 ± 5.7), was significantly lower compared to the PR-Cog+ and PR–Cog– groups (10.8 ± 7.1, *p* = 0.014 and 9.7 ± 7.2, *p* = 0.012, respectively). As expected, the robust group had lowest number of prescribed medications and the lowest number of comorbidities compared to other groups. However, these variables could only distinguish the cognitively frail group from the PR+Cog+ group and not from the PR–Cog+ group. Interestingly, concern about falls is not a predictor of cognitive frailty (*p* > 0.050). Overall, among basic demographics and clinical characteristics without MMSE, BMI had the largest effect size in distinguishing the cognitive frailty group from other groups.

### 3.2. Association Between Cognitive Impairment and Sensor-Derived Parameters

Among the assessed nocturnal sleep parameters, TiB (rho = 0.24, *p* = 0.002), TST (rho = 0.25, *p* = 0.001), and side-lying (rho = 0.21, *p* = 0.007) showed a significant positive correlation with MMSE (Table 2). All sensor-derived physical activity parameters showed a significant correlation with MMSE. Physical activity parameters such as sedentary duration were negatively associated with MMSE (rho = –0.29, *p* < 0.001), while light activity (rho = 0.24, *p* = 0.003), MtV activity (rho = 0.29, *p* < 0.001), and number of steps (rho = 0.33, *p* < 0.001) had positive correlations (Table 2).

### 3.3. Comparison of Aging Groups

Table 3 summarizes sensor-derived sleep and physical activity parameters for each group. The PR+Cog+ group had the highest TiB (8.2 ± 2.0 h) and TST (6.1 ± 1.5 h) values, whereas the PR–Cog– group had the lowest TiB (6.4 ± 2.1) and TST (4.6 ± 1.9) values. The cognitive frailty group had the highest SOL (19.7 ± 8.5) and highest prone sleep position percentage (19.3% ± 20.4), while the PR+Cog+ group had the highest side-lying sleep position percentage (33.8%). Overall, among assessed sleep parameters, TiB had the largest effect size for distinguishing the cognitive frailty group from the PR+Cog+ group (ES = 0.91), and the side-lying position had the largest effect size in distinguishing the cognitive frailty group from the PR-Cog+ group (ES = 0.55). Figure 2 illustrates the prevalence of cases with TST exceeding 6 h. The prevalence of TST > 6 h was 32% in the cognitive frailty group, which was significantly lower compared to the PR-Cog+ (63%, *p* = 0.008) and PR+Cog+ (78%, *p* < 0.001) groups.

Among sensor-derived physical activity parameters, the cognitive frailty group had the highest proportion of sedentary behavior (85.9 ± 6.4%) and the lowest proportion of light activity (12.9 ± 5.4%) compared to other groups (*p* < 0.050). The cognitive frailty group also had the lowest proportion of MtV activity (1.3 ± 1.6%), but the difference achieved statistical significance only when compared with the PR+Cog+ group (*p* < 0.001). Among assessed postures, only percentage of walking distinguished the cognitive frailty group from other groups, whereas standing distinguished between the cognitive frailty and PR+Cog+ groups. Results also suggest that number of steps over 48 h distinguishes between cognitive frailty groups and other groups (*p* < 0.002).

Figure 3 illustrates some of the top sensor-derived parameters selected among sleep, activity behaviors, and cumulative postures and locomotion, which had relatively large effect sizes in distinguishing the cognitive frailty group from other groups. Overall, among sensor-derived parameters, the largest effect size was observed in percentage of time walking per day (ES = 1.87) and daily step count (ES = 0.86), respectively.

The top six independent predictors (sedentary behavior, %; MtV activity, min; MtV activity, %; standing, %; walking, %; steps, number) in the decision tree model identified the PR-Cog- group with the following performance: sensitivity= 0.93 (95%CI: 0.88–0.98), specificity= 0.57 (95%CI: 0.35–0.79), accuracy= 0.86 (95%CI: 0.81–0.90), and AUC = 0.75 (95%CI: 0.64–0.85) (Figure 4).

## 4. Discussion

In this study we examined whether everyday activity parameters (e.g., cumulative postures, activity behavior, locomotion, and nocturnal sleep quantity) derived from a single chest-worn sensor could identify older adults with cognitive frailty. We demonstrated that six independent sensor-derived parameters representing cumulative postures (duration of standing and walking at percentage over 24 h), activity behavior (percentage of sedentary behavior and MtV activity over 24 h), and locomotion (total number of steps over 48 h) collectively distinguish older adults with cognitive frailty from age-matched older adults without cognitive frailty. These results may support the application of wearables to remotely monitor risk of cognitive frailty through physical activity monitoring.

A growing number of studies suggest that motoric cognitive risk syndrome or cognitive frailty is associated with high risks of cognitive impairment, dementia, and other adverse health outcomes [34,63]. However, cognitive frailty assessment is typically operationalized at one time point. Our study may open the opportunity to provide data longitudinally rather than with a one-time assessment and thus better identify individuals who develop cognitive frailty or associated adverse health outcomes such as dementia and mortality outcomes. While we used a chest-worn sensor to extract everyday motoric cognitive risk parameters, we believe that any other sensor type could extract similar parameters could benefit from the proposed model to identify cognitive frailty. Our proposed method could be also deployed in remote patient monitoring devices to allow tracking of changes in motoric cognitive risk over time.

The high amount of sedentary behavior and negligible walking percentage in the cognitive frailty group confirm that the co-occurrence of frailty and cognitive impairment likely exacerbates the frailty cascade of negative energy balance, sarcopenia, diminished strength, and exertion intolerance [64]. To the best of our knowledge, prior studies have not reported on the ability to identify older adults with cognitive frailty using sensor derived parameters alone.

Our results suggest that the combined prevalence of cognitive impairment and physical frailty increases with the prevalence of sleep disruption. This is aligned with previous studies suggesting that poor sleep is a common indicator of both cognitive impairment [65,66] and physical frailty [17]. An interesting sleep marker observed in our study was the low duration of side-lying sleep position among participants with both cognitive impairment and physical frailty. These findings build on our prior work demonstrating that a lower side sleep position duration is associated with a higher likelihood of falls in an acute setting [45].

The decision tree demonstrated a sensitivity of 0.93 for detecting cognitive frailty in older adults, while its specificity was 0.57. In other words, the classifier catches the actual cases of cognitive frailty, but it also has a high false positive rate. It is recommended that screening tests be designed with high sensitivity to avoid missing any true positive cases [67]. Therefore, the proposed model would be a preclinical screening test to triage those who could benefit from further clinical examination to confirm cognitive frailty syndrome.

In this study, we used a pendant sensor to collect parameters of interest. The advantage of the pendant sensor form is that it can be worn unobtrusively as a necklace, which may improve its wearability over long periods of time. This is an important requirement for longitudinal and remote monitoring applications [44,68]. On the other hand, a pendant or chest-worn sensor form may have limited accuracy and/or may provide limited information about activities of daily living and sleep parameters of interest. The sensor used in this study used a comfortable lanyard with magnetic closure to maximize the level of comfort and reduce potential adverse events such as choking while sleeping (e.g., the lanyard with magnetic closure quickly disconnects to prevent any harmful pressure on the user’s neck). In addition, the ergonomics of the sensor were designed to reduce artifact movements, which may reduce accuracy (e.g., flipping of the sensor, not following movement of the chest, etc. [41,42]). All participants in this study self-reported high acceptability and level of comfort for the sensor, and no adverse events were reported.

Improving the level of comfort may decrease accuracy when measuring physical activity parameters of interest under unsupervised conditions, for instance, when the sensor is not worn under a shirt, leading to signal artifacts (e.g., the sensor may not necessary follow the chest movements), and when there is a chance of flipping of the sensor (e.g., uncertainty in identifying sensor orientation essential for postural transition detection and classification [41,42]). Securing the sensor on the skin [45] or integrating it into a tank top shirt, as suggested in our previous study [54], may address these limitations, but it may also reduce its acceptability for wear over longer periods of time. Networks of sensors can provide elaborate information about older adults’ daily physical activity of the older adults and thus improve accuracy in identifying cognitive frailty syndrome. On the other hand, an increase in the number of sensors may reduce level of comfort and acceptability, increase cost and analysis burden [69]. The optimal choice of sensor type, sensor attachment location, number of needed sensors, sensor ergonomic design or method of wearing, and accuracy for extracting parameters of interest are outside of scope of the current study and should be addressed by future studies.


**Limitations**


This study has several limitations. Our sample size (*N* = 159) may be underpowered for a robust between-groups comparison for some of the parameters of interest. To identify motoric cognitive risk syndrome, we used a more conservative cut-point for both determination of cognitive risk and physical frailty risk. This approach may not be conventional for identifying those with clinically confirmed cognitive impairment or physical frailty, but it enables balancing the sample size in each group. While unconventional, these cut-points are supported by previous literatures as indicators of those with risk of motoric cognitive risk syndrome [38,39]. In this study, we had fewer participants with fit physical fitness and impaired cognition (PR+Cog-), *n* = 4. Therefore, we removed them from group comparison. This is aligned with other studies suggesting low prevalence of those with only cognitive impairment and no physical frailty [63]. A study with a larger sample size is required to investigate the ability of sensor-derived parameters to distinguish the cognitive frailty group from the PR+Cog- group.

As discussed above, this study used a pendant sensor to derive physical activity and sleep parameters of interest. The proposed form factor may, however, be inaccurate or insufficient to extract all physical activity- or sleep-related parameters of interest. Finally, the duration of physical activity monitoring was 48 h, which may have limited reliability and may not be sufficient time to represent every day physical activity patterns of community-living older adults. However, as discussed in a previous study, two consecutive days could be sufficient to extract some of daily physical activity patterns such as cumulative postures and postural transition data [52]. On the other hand, a high time resolution (e.g., 48 h) to determine cognitive frailty could facilitate tracking subtle changes in cognitive frailty status over time, which in turn could be used to identify the onset of modifiable risk factors for cognitive frailty, such as effect of medication and other health problems (e.g., sleep deprivation, unmanaged pain, depression, etc.). Another study should be performed to examine the reliability of the proposed model using 48 h of monitoring to identify cognitive frailty cases.

## 5. Conclusions

We demonstrated the feasibility of using a single chest-worn sensor to monitor sleep, physical activity, and postural activities in an older community-living population. Sensor-derived data were readily transformed into digital markers that can identify older adults with motoric cognitive risk syndrome or cognitive frailty. With further testing and validation, the sensor parameters measured in our model can facilitate remote patient monitoring to determine and track cognitive frailty over time using wearable sensors, especially when connected to home technologies. With reliable data on everyday activities, clinicians may quickly identify those at risk, potentially identify modifiable risk factors for cognitive frailty, and better align treatment options that target physical frailty and cognitive impairment risks. Therefore, these applications (connected homes with linked wearable sensors) hold the promise of improving functional independence and reducing health care costs by delaying transitions to long-term care among at-risk older adults.

## Figures and Tables

**Figure 1 sensors-20-02218-f001:**
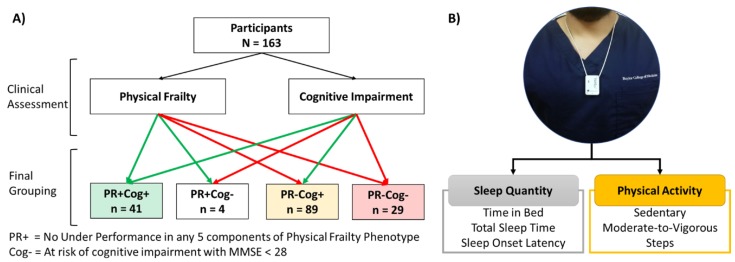
(**A**) CONSORT diagram of participants recruitment. (**B**) Placement of sensor and the samples of variables extracted from a single sensor.

**Figure 2 sensors-20-02218-f002:**
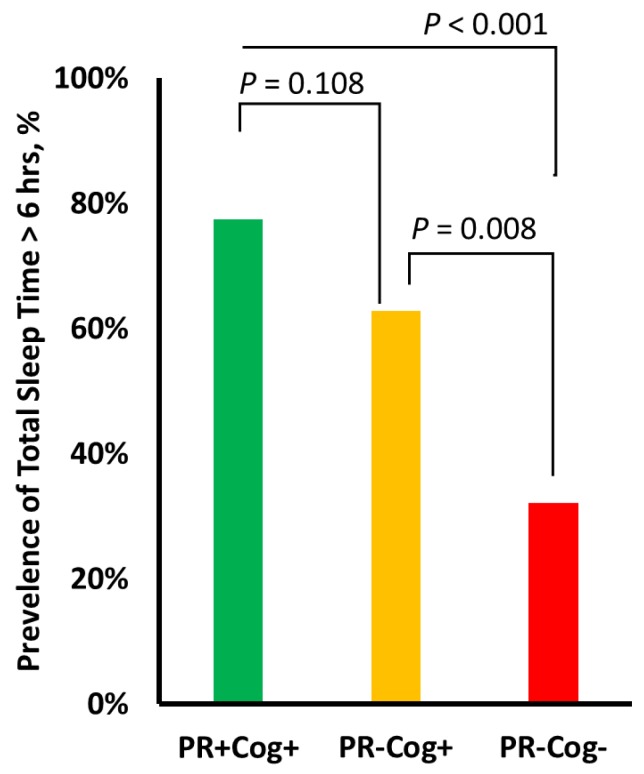
The prevalence of cases with total sleep time exceeding 6 h per night in each of the three groups: physically robust and high cognitive function (PR+Cog+), signs of physical frailty with high cognitive function (PR-Cog+), and signs of physical frailty with risk of cognitive impairment (PR-Cog-), defined as cognitive frailty in this study. Lowest significant sleep deprivation was observed in the cognitive frailty group.

**Figure 3 sensors-20-02218-f003:**
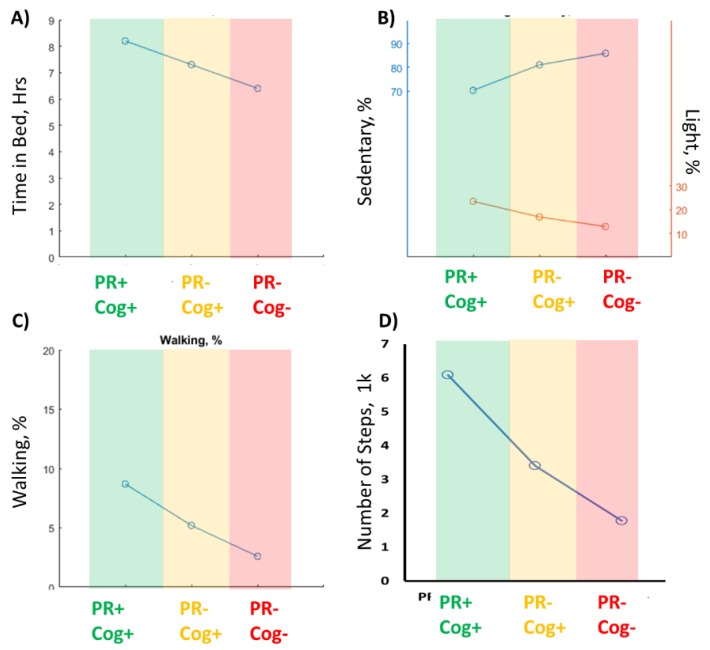
The top 4 sensor-derived parameters to identify the cognitive frailty group (PR-Cog-) from other groups. (**A**) the time in bed declined from robust group (PR+Cog+) to cognitive frailty group. (**B**) The percentage of sedentary duration increased while light activity reduced from Robust to cognitive frailty group. (**C**) the percentage of walking duration declined from Robust group to cognitive frailty group. (**D**) The number of steps reduced from robust group to cognitive frailty group. .

**Figure 4 sensors-20-02218-f004:**
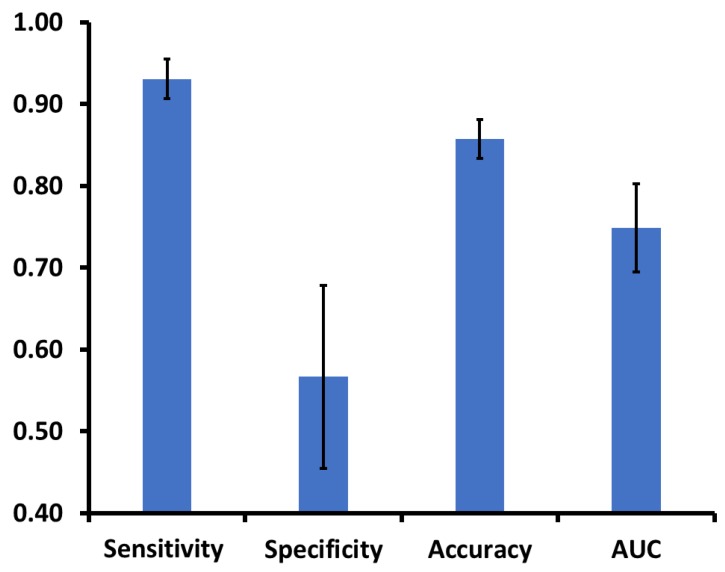
The classifier (decision tree) distinguishes older adults with cognitive frailty (PR-Cog-) from the rest of the aging groups. The classifier includes sensor-derived parameters and has an area under the curve greater than 0.75 on average.

**Table 1 sensors-20-02218-t001:** Demographic and clinical characteristics reported by mean ± standard deviation.

	Physically Robust (PR+Cog+) *n* = 41	Not Physically Robust (PR-)		PR+Cog+ Vs PR-Cog+	PR+Cog+ Vs PR-Cog-	PR-Cog+ Vs PR-Cog-
		High Cognitive Performance (PR-Cog+) *n* = 89	Cognitive Impairment Risk(PR-Cog-) *n* = 29			
	mean ± SD	mean ± SD	mean ± SD	*p*-Value (Effect Size)		
Age, years	73.4 ± 7.2	76.5 ± 11.7	74.7 ± 10.8	0.123(0.32)	0.626(0.14)	0.416(0.16)
BMI, kg/m^2^	24.9 ± 5.7	29.7 ± 7.0	31.7 ± 6.6	0.000(0.75)	0.000(1.09)	0.170(0.29)
MMSE	29.5 ± 0.7	29.2 ± 0.8	24.5 ± 2.6	0.303(0.34)	0.000(2.65)	0.000(2.49)
Concern about falls (FES-I)	20.2 ± 3.6	26.1 ± 13.4	22.9 ± 11.4	0.007(0.59)	0.326(0.32)	0.199(0.25)
Depression (CES-D)	6.5 ± 5.7	9.7 ± 7.2	10.8 ± 7.1	0.014(0.49)	0.012(0.65)	0.477(0.15)
Medications	2.4 ± 1.8	4.6 ± 3.6	4.6 ± 4.9	0.002(0.77)	0.034(0.60)	0.968(0.01)
Comorbidities	2.3 ± 1.8	3.9 ± 2.1	4.3 ± 2.1	0.000(0.81)	0.002(1.03)	0.519(0.19)

BMI = body mass index, CES-D = Center for Epidemiologic Studies Depression Scale, MMSE = Mini-Mental State Examination, FES-I = Falls Efficacy Scale International. PR+ = physically robust, PR- = Presence of any frailty phenotype, Cog+ = high cognitive performance, Cog- = cognitive impairment risk.

**Table 2 sensors-20-02218-t002:** Correlation between sensor-derived parameters and the cognitive status of participants measured by Mini-Mental State Examination (MMSE).

Correlation, rho (*p*-Value) ^†^	
**Sleep Parameters**	
Time in Bed, h	0.24(0.002) *
Sleep Onset Latency, min	−0.10(0.201)
Total Sleep Time, h	0.26(0.001) *
Sleep Efficiency, %	0.12(0.129)
Sleep Supine, %	0.05(0.489)
Sleep Prone, %	−0.10(0.095)
Sleep Sides, %	0.21(0.007) *
**Physical Activity**	
Sedentary, h	−0.29(0.000) *
Sedentary, %	−0.30(0.000) *
Light, h	0.24(0.003) *
Light, %	0.28(0.000) *
Moderate-to-Vigorous, min	0.29(0.000) *
Moderate-to-Vigorous, %	0.27(0.001) *
Sitting, %	0.01(0.829)
Standing, %	0.19(0.020) *
Walking, %	0.34(0.000) *
Lying, %	−0.10(0.050) *
Number of Steps, 1K	0.33(0.000) *

^†^ = measured by Spearman Correlation coefficients; * = the parameters with *p*-value < 0.050; 1K = 1000.

**Table 3 sensors-20-02218-t003:** Mean and standard deviation of sensor-derived parameters and comparison of the four aging trajectories.

	Physically Robust (PR+Cog+)	Signs of Frailty Phenotype (PR-)		PR+ Cog+ Vs PR-Cog+	PR+Cog+ Vs PR-Cog-	PR-Cog+ Vs PR-Cog-
		High Cognitive performance (PR-Cog+)	Cognitive Impairment Risk (PR-Cog-)			
	Mean ± SD			*p*-Value (Effect Size)		
**Sleep Parameters**						
Time in Bed, h*	8.2 ± 2.0	7.3 ± 2.1	6.4 ± 2.1	0.023(0.45)	0.000(0.91)	0.037(0.45)
Total Sleep Time, h	6.1 ± 1.5	5.5 ± 1.9	4.6 ± 1.9	0.082(0.36)	0.001(0.88)	0.024(0.47)
Sleep Onset Latency, min	16.8 ± 7.7	18.7 ± 8.0	19.7 ± 8.5	0.227(0.24)	0.144(0.36)	0.551(0.13)
Wake After Sleep Onset, h	1.7 ± 0.8	1.4 ± 0.7	1.4 ± 0.7	0.005(0.55)	0.004(0.72)	0.438(0.17)
Sleep Efficiency, %	78.1 ± 9.3	78.6 ± 10.8	76.5 ± 12.0	0.818(0.05)	0.548(0.15)	0.378(0.18)
Sleep Supine, %	45.1 ± 20.3	42.6 ± 26.4	44.0 ± 24.4	0.593(0.11)	0.855(0.05)	0.792(0.06)
Sleep Prone, %	13.7 ± 17.3	12.4 ± 18.2	19.3 ± 20.4	0.712(0.07)	0.222(0.29)	0.088(0.36)
Sleep Sides, %	33.8 ± 17.3	33.6 ± 23.9	20.8 ± 22.9	0.952(0.01)	0.018(0.64)	0.009(0.55)
**Physical Activity Parameters**						
Sedentary, h	9.5 ± 2.6	11.9 ± 3.8	12.9 ± 2.7	0.000(0.73)	0.000(1.29)	0.146(0.32)
Sedentary, %*	70.3 ± 12.9	81.0 ± 8.9	85.9 ± 6.4	0.000(0.96)	0.000(1.52)	0.022(0.64)
Light, h	3.2 ± 1.3	2.4 ± 1.2	1.9 ± 0.9	0.002(0.57)	0.000(1.10)	0.045(0.49)
Light, %*	23.5 ± 10.0	16.9 ± 7.7	12.9 ± 5.4	0.000(0.75)	0.000(1.33)	0.024(0.60)
Moderate-to-Vigorous, min	49.3 ± 31.6	19.2 ± 20.5	11.2 ± 14.1	0.000(1.13)	0.000(1.56)	0.116(0.45)
Moderate-to-Vigorous, %	6.1 ± 4.1	2.2 ± 2.3	1.3 ± 1.6	0.000(1.17)	0.000(1.55)	0.155(0.45)
Sitting, %	44.1 ± 15.7	47.5 ± 16.4	43.8 ± 18.9	0.287(0.21)	0.951(0.01)	0.314(0.21)
Standing, %	16.8 ± 5.9	13.4 ± 6.0	11.5 ± 5.0	0.003(0.56)	0.000(0.96)	0.133(0.35)
Walking, %*	8.7 ± 4.0	5.2 ± 3.4	2.6 ± 2.3	0.000(0.95)	0.000(1.87)	0.001(0.91)
Lying, %	30.3 ± 16.0	33.8 ± 19.9	42.1 ± 21.8	0.352(0.19)	0.015(0.61)	0.052(0.40)
Number of Steps, 1K *	6.1 ± 3.1	3.4 ± 2.2	1.8 ± 1.6	0.000(0.99)	0.000(1.74)	0.002(0.86)

PR+ = physically robust, PR- = presence of any frailty phenotype, Cog+ = high cognitive performance, Cog- = cognitive impairment risk. SD = standard deviation. h= hours, min= minutes, 1K = 1000, % = percentage, TST = total sleep time. * Number of steps were reported over 48 h monitoring.

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
