# Peer review of "Toward Using Wearables to Remotely Monitor Cognitive Frailty in Community-Living Older Adults: An Observational Study"

_sensors, 2020, doi:10.3390/s20082218_

Round 1

Reviewer 1 Report

This paper aims to classify patients into four aging groups, focusing on the identification of cognitive frailty. It proposes the use of commercial wearable sensors for this task.

The paper asserts that the contribution is identification into the 4 give groups, however the clinical relevance of this is not sufficiently argued.

Overall the paper’s aim is hard to understand – it is not a new technology, a new use of sensors or a new algorithm. On the clinical side the relevance and clinical implications are not discussed.

In general, I disagree with the first assumption of the paper – that a pendant wearable sensor is accurate enough to measure gait and sleep parameters. To convince a reader of this there needs to be literature which shows this system, as a pendant, in a study with older adults where the sleep and motion parameters are validated against a gold standard. This is needed in order to understand the accuracy and reproducibility of the results.

In my opinion, the only way to prove that the technological choice was valid for such a task would be to repeat the experiment qand prove reproducibility.

Regarding the decision tree – this was not discussed, why a decision tree, why not other algorithms? How were the parameters and thresholds selected? What was the test-train-validate split?

The limitation of only assessing for 48 consecutive hours is a major limitation, as it only gives and indication of the patients state on these two days.

When were the clinical assessments carried out, how large was the time gap between assessment and use of the sensor? As both are only snap shots, they need to be very close together.

Who performed the clinical assessments?

Please report the number of technical problems and failures.

Is a pendant worn when sleeping not a choking hazard, and as it is not attached to the body, how is it actually measuring the activity and not the interaction of the pendant and the body.

Some formatting issues :lack of space before references, too large spaces, double full stops. English grammar and sentence structure needs to be improved.

Please add the full details of the ethics approval (e.g. identification number and full name of the committee).

Parameters such as moderate to vigorous activity and sleep efficiency need to be defined as there is no universal definition of these terms.

Author Response

Q1: This paper aims to classify patients into four aging groups, focusing on the identification of cognitive frailty. It proposes the use of commercial wearable sensors for this task. The paper asserts that the contribution is identification into the 4 groups, however the clinical relevance of this is not sufficiently argued.

A1: We amended the first paragraph of the introduction section to better describe the clinical relevance.

Q2: Overall the paper’s aim is hard to understand – it is not a new technology, a new use of sensors or a new algorithm. On the clinical side the relevance and clinical implications are not discussed.

A2: We appreciate the scientific comments from the reviewer. As the reviewer recognized, we are utilizing the body worn sensor to remotely determine those who may suffer from cognitive frailty, which is known to be an important predictor for cognitive decline over time and risk of dementia. This aim is aligned with the scope of the Sensors Special Issue on Body Worn Sensors and Related Applications- https://www.mdpi.com/journal/sensors/special_issues/body_worn_sensors). As specified by this Call for papers, some of the examples fit to this special issue as “new studies that utilize sensors to extract digital biomarkers associated with cognitive decline, motor capacity deterioration because of specific conditions (e.g., dementia, Parkinson’s disease, stroke, cancer, diabetes), …”. Our study proposed the use of wearable sensors to extract digital biomarkers that serve as surrogate markers of cognitive frailty in older adults. We rephrased the title of the manuscript and significantly revised the introduction section to better highlight the focus of our study and current gaps in monitoring cognitive frailty. We recognize that the focus of this paper is not technical development but rather exploring clinically meaningful measures of cognitive frailty using a wearable sensor. To the best of our knowledge, this is the first study suggesting the use of remote activity monitoring to measure cognitive-frailty among older at-risk adults.  

Q3: In general, I disagree with the first assumption of the paper – that a pendant wearable sensor is accurate enough to measure gait and sleep parameters. To convince a reader of this there needs to be literature which shows this system, as a pendant, in a study with older adults where the sleep and motion parameters are validated against a gold standard. This is needed in order to understand the accuracy and reproducibility of the results.

A3: The estimation of accuracy of the pendant sensor to measure gait and sleep is beyond the focus of the current study. We therefore reference prior studies that validate the accuracy of wearable sensors in this context. Our study suggests that physical activities, including metrics associated with gait and sleep extracted from a pendant sensor, could be used to distinguish those with cognitive frailty. We recognize that a pendant sensor may have limitations to accurately quantify daily physical activities, including sleep and gait patterns. We acknowledge this in the Limitation section of the Discussion.

Q4: In my opinion, the only way to prove that the technological choice was valid for such a task would be to repeat the experiment and prove reproducibility.

A4: We appreciate the scientific comments from the review. As described in response to Q3, the focus of this study is not to demonstrate the validity of wearable sensors as measures of physical activities or sleep. In the Limitations section we highlighted that additional studies with larger numbers of older adults are warranted to confirm the findings of this study. We highlight that the use of an algorithm or sensor form factor may have limitations for measuring physical activities and/or sleep parameters of interest (page 13, lines 333-356).

Q5: Regarding the decision tree – this was not discussed, why a decision tree, why not other algorithms? How were the parameters and thresholds selected? What was the test-train-validate split?

A5: In medical decision making, conceptually simple decision-making models with the possibility of automatic learning are the most appropriate. Decision trees are a reliable and effective decision-making technique that provide high classification accuracy with simple representation of gathered knowledge. They have been used in different areas of medical decision making [60]. We used MATLAB software to develop the model and chose the default value for “maximal number of decision splits,” “leaf merge flag,” “maximum tree depth,” and “Number of bins for numeric predictors.”  We also added an additional description on the K-fold cross validation (page 3, lines 179-189).

Q6: The limitation of only assessing for 48 consecutive hours is a major limitation, as it only gives an indication of the patients’ state on these two days.

A6: We agree with the reviewer that prolonged recording will provide better understanding of the physical activity pattern of older adults and may better capture variation. At the same time, the goal is having a system with sufficient time resolution to track subtle changes in cognitive frailty over time caused by medical management factors such as medications (e.g., sleep medication, antidepressant, etc.), and unmanaged symptoms (e.g., pain) for timely intervention. Clinicians prefer a fast, accurate measure, and the lengthy recording may not support a clinically timely “when-to-treat” strategy. We revised the Methods (Methods, section 2.3; page 4). We have also acknowledged in the Limitation section that two days of monitoring physical activity may have limited reliability to evaluate cognitive frailty; thus, further replication studies should be conducted to evaluate the reliability of two-day monitoring to determine cognitive frailty (page 13).

Q7: When were the clinical assessments carried out, how large was the time gap between assessment and use of the sensor? As both are only snap shots, they need to be very close together.

A7: All clinical assessments were performed at the time of the clinical visit. The sensor was given to patients at the time of visits, and the data was recorded for two days. The patient returned the sensor by shipping it back in a prepaid envelope or giving it to a study coordinator at the next visit. Methods section 2.3 is revised to reflect these details.

Q8: Who performed the clinical assessments?

A8: The clinical assessments were performed by trained research assistants. We added the names of research assistants who assisted with analyzing the sensor data in the acknowledgement section. The Methods section 2.2 is revised to reflect these edits.

Q9: Please report the number of technical problems and failures.

A9: We didn’t have any noticeable technical difficulties to report. However, it may be possible that a subject did not wear the sensor for the entire 48-hour period. We have previously demonstrated the high degree of adherence to continuously wearing a chest-worn sensor for 48 hours (Najafi et al, Diabetes Care, 2010), as well as self-reported adherence to wearing the sensor. We added this information to the Limitation section.

Q10: Is a pendant worn when sleeping not a choking hazard, and as it is not attached to the body, how is it actually measuring the activity and not the interaction of the pendant and the body.

A10: We anticipate the risk is minimal for wearing the sensor, which is a lanyard necklace with a magnetic closure that will be detached if too much force is applied. No discomfort or adverse events were reported in our study. Again, we would like to emphasize that our goal is not to promote a pedant sensor or any specific sensor brand but rather to describe physical and sleep-derived metrics that could be used to remotely monitor cognitive-frailty. We hope this research encourages implementation of this method in future activity monitoring platforms which enable measuring similar metrics to determine and track subtle changes in cognitive frailty. We revised the Conclusion section to better highlight this implication. (page 13, lines 350-356)

Q11: Some formatting issues: lack of space before references, too large spaces, double full stops. English grammar and sentence structure needs to be improved.

A11: We have edited this manuscript for formatting, grammar, and clarity.

Q12: Please add the full details of the ethics approval (e.g. identification number and full name of the committee).

A12: The Methods section has been revised with these details (Section 2.1).

Q13: Parameters such as moderate to vigorous activity and sleep efficiency need to be defined as there is no universal definition of these terms.

A13: We revised the Methods section to add the definition of variables. The sleep parameters derived from the sensor are defined by American Academy of Sleep Medicine. (page 4, lines 151-168)

Reviewer 2 Report

The authors presented an interesting study to examine whether remote physical activity (PA) monitoring could be used to distinguish those with cognitive frailty.

To this reviewer, the topic is interesting and the manuscript was well-written. However there are some minor limitations. I suggest the authors to address these before publishing their inetersting study:

1-    Minor editing is needed to improve the presentation, for example see lines 58, 197, 205, 207.

2-    Line 159-160: I suggest to move the first sentence to the end of paragraph.

3-    Line 126: The authors should justify why they used this type of sensor. There are currently many wearable sensors for monitoring ADL. They need to review them and justify their choice. I suggest them to review and cite these articles:

·       Esfahani, M. I. M., & Nussbaum, M. A. (2018). A “Smart” undershirt for tracking upper body motions: task classification and angle estimation. IEEE Sensors Journal, 18(18), 7650-7658.

·       Mukhopadhyay, Subhas Chandra. "Wearable sensors for human activity monitoring: A review." IEEE sensors journal 15.3 (2015): 1321-1330.

·       Incel, Ozlem Durmaz, Mustafa Kose, and Cem Ersoy. "A review and taxonomy of activity recognition on mobile phones." BioNanoScience 3.2 (2013): 145-171.

·       Shoaib, Muhammad, et al. "A survey of online activity recognition using mobile phones." Sensors 15.1 (2015): 2059-2085.

·       Schrack, Jennifer A., et al. "Assessing daily physical activity in older adults: unraveling the complexity of monitors, measures, and methods." Journals of Gerontology Series A: Biomedical Sciences and Medical Sciences 71.8 (2016): 1039-1048.

·       Mokhlespour Esfahani, Mohammad Iman, and Maury A. Nussbaum. "Classifying Diverse Physical Activities Using “Smart Garments”." Sensors 19.14 (2019): 3133.

4-    The placement of sensor on the body for detecting the ADL is questionable. They need to justify why they used this placement especially a single chest-worn sensor may not be sufficient for monitoring the ADL. I encourage the authors to read the recent usability study for wearable sensors for monitoring the activities and cite them:

·       Cleland, Ian, et al. "Optimal placement of accelerometers for the detection of everyday activities." Sensors 13.7 (2013): 9183-9200.

·       Mokhlespour Esfahani, Mohammad, and Maury Nussbaum. "Preferred placement and usability of a smart textile system vs. inertial measurement units for activity monitoring." Sensors 18.8 (2018): 2501.

·       Boerema, Simone T., et al. "Optimal sensor placement for measuring physical activity with a 3D accelerometer." Sensors 14.2 (2014): 3188-3206.

·       Özdemir, Ahmet Turan. "An analysis on sensor locations of the human body for wearable fall detection devices: Principles and practice." Sensors 16.8 (2016): 1161.

5-    Also the evaluation of their method is not clear about PR+Cog- group since they removed this group from their comparison because of limited number of participants. This limitation should be explained in discussion.

Author Response

Q14: The authors presented an interesting study to examine whether remote physical activity (PA) monitoring could be used to distinguish those with cognitive frailty. To this reviewer, the topic is interesting and the manuscript was well-written. However there are some minor limitations. I suggest the authors to address these before publishing their inetersting study:

A14: we appreciate the constructive comments from the reviewer. Cognitive frailty has attained significant attention as an emerging clinical challenge in the older adult population.

Q15: Minor editing is needed to improve the presentation, for example see lines 58, 197, 205, 207.

A15: We have significantly revised the introduction section to improve the presentation. Other sections were also revised to improve clarity.

Q17: Line 159-160: I suggest to move the first sentence to the end of paragraph.

A17: This change has been made.

Q18: Line 126: The authors should justify why they used this type of sensor. There are currently many wearable sensors for monitoring ADL. They need to review them and justify their choice. I suggest them to review and cite these articles:

  • Esfahani, M. I. M., & Nussbaum, M. A. (2018). A “Smart” undershirt for tracking upper body motions: task classification and angle estimation. IEEE Sensors Journal, 18(18), 7650-7658.
  • Mukhopadhyay, Subhas Chandra. "Wearable sensors for human activity monitoring: A review." IEEE sensors journal 15.3 (2015): 1321-1330.
  • Incel, Ozlem Durmaz, Mustafa Kose, and Cem Ersoy. "A review and taxonomy of activity recognition on mobile phones." BioNanoScience 3.2 (2013): 145-171.
  • Shoaib, Muhammad, et al. "A survey of online activity recognition using mobile phones." Sensors 15.1 (2015): 2059-2085.
  • Schrack, Jennifer A., et al. "Assessing daily physical activity in older adults: unraveling the complexity of monitors, measures, and methods." Journals of Gerontology Series A: Biomedical Sciences and Medical Sciences 71.8 (2016): 1039-1048.
  • Mokhlespour Esfahani, Mohammad Iman, and Maury A. Nussbaum. "Classifying Diverse Physical Activities Using “Smart Garments”." Sensors 19.14 (2019): 3133.

A18: We appreciate the reviewer’s comments, and we have cited these references. The choice of sensor type is outside the scope of this study.

Q19: The placement of sensor on the body for detecting the ADL is questionable. They need to justify why they used this placement especially a single chest-worn sensor may not be sufficient for monitoring the ADL. I encourage the authors to read the recent usability study for wearable sensors for monitoring the activities and cite them:

  • Cleland, Ian, et al. "Optimal placement of accelerometers for the detection of everyday activities." Sensors 13.7 (2013): 9183-9200.
  • Mokhlespour Esfahani, Mohammad, and Maury Nussbaum. "Preferred placement and usability of a smart textile system vs. inertial measurement units for activity monitoring." Sensors 18.8 (2018): 2501.
  • Boerema, Simone T., et al. "Optimal sensor placement for measuring physical activity with a 3D accelerometer." Sensors 14.2 (2014): 3188-3206.
  • Özdemir, Ahmet Turan. "An analysis on sensor locations of the human body for wearable fall detection devices: Principles and practice." Sensors 16.8 (2016): 1161.

A19: We appreciate the reviewer’s comment. The placement of the sensor is outside the scope of this study, but we used a validated algorithm to extract sensor-derived parameters. The focus of this study is to examine whether sensor-derived metrics, i.e. indicators of physical activities and sleep, could be used to determine the presence of cognitive-frailty. We hope our study encourages implementation of cognitive-frailty monitoring in future wearable devices, which would enable measuring of physical activity and sleep parameters of interest. We also hope to encourage more research in the area of remote monitoring of cognitive frailty, which is known to be an important but potentially modifiable geriatric risk factor.  

Q20: Also the evaluation of their method is not clear about PR+Cog- group since they removed this group from their comparison because of limited number of participants. This limitation should be explained in discussion.

A20: We revised the Limitations paragraph in the Discussion section to include this information. (page 13, lines 333-356)

Reviewer 3 Report

Authors evaluated the feasibility to use remote physical activity monitory to distinguish people with cognitive-frailty. Manuscript is well-written and the proposed topic is interesting. The applied methodology is properly designed for the application and the results are well presented. 

However, I would like to provide some suggestions to improve the scientific quality of the paper and increase its readability.

Title: I suggest to be more specific avoiding the generalization with "wearable sensors".

Paragraph 2 was missing, I suggested to insert 2 Materials and Methods before the subparagraph 2.1.

Paragraph 2.3: please provide more details on the specifications of the pendant sensors (accuracy, resolution, range etc)

Paragraph 2.4: please provide more details on the decision tree classifier. How many splits were allowed? Which was the criterion for the split? How was the training phase performed, supervised? unsupervised?

Paragraph 2.4: I suggest to compute also the precision and F1-score to have a more complete overview on the classifier performance (see Taborri et al. https://doi.org/10.3390/s19061461). In addition, the equation of all computed parameters should be reported.

Paragraph 2.4:  I suggest to insert a range of AUC values to understand the performance of classifier; Authors could use the range reported for the G-index in the previous suggested paper.

Discussion: Discussions related to the machine-learning approach are poor. Please enrich for example by discussing why the specificity is lower than sensitivity and which are the possible consequences.

Author Response

Q21: Authors evaluated the feasibility to use remote physical activity monitory to distinguish people with cognitive-frailty. Manuscript is well-written and the proposed topic is interesting. The applied methodology is properly designed for the application and the results are well presented. However, I would like to provide some suggestions to improve the scientific quality of the paper and increase its readability.

A21: We appreciate the suggestions from the reviewer to further enhance the manuscript.

Q22: Title: I suggest to be more specific avoiding the generalization with "wearable sensors".

A22: We revised the title to: “Toward Using Wearable to Remotely Monitor Cognitive Frailty in Community-Living Older Adults: an Observational Study”

Q23: Paragraph 2 was missing, I suggested to insert 2 Materials and Methods before the subparagraph 2.1.

A23: We revised the manuscript format to include this change.

Q24: Paragraph 2.3: please provide more details on the specifications of the pendant sensors (accuracy, resolution, range etc)

A24: The description of the sensor has been added to the Methods section.

Q25: Paragraph 2.4: please provide more details on the decision tree classifier. How many splits were allowed? Which was the criterion for the split? How was the training phase performed, supervised? unsupervised?

A25: We revised manuscript to address these questions. Please also refer to A5.

Q26: Paragraph 2.4: I suggest to compute also the precision and F1-score to have a more complete overview on the classifier performance (see Taborri et al. https://doi.org/10.3390/s19061461). In addition, the equation of all computed parameters should be reported.

A26: We revised the Methods section to include this information (page 5, lines 186-189).

Q27: Paragraph 2.4: I suggest to insert a range of AUC values to understand the performance of classifier; Authors could use the range reported for the G-index in the previous suggested paper.

A27: We have added the 95%CI to the Results section 3.3. (page 10, lines 261-262)

Q28: Discussion: Discussions related to the machine-learning approach are poor. Please enrich for example by discussing why the specificity is lower than sensitivity and which are the possible consequences.

A28: We have amended the Discussion section to include the benefit of high sensitivity in our findings. (page 12, lines 301-307).

Round 2

Reviewer 1 Report

Thank you for your significant work on improving the manuscript.

I feel that it now provides enough detail and discussions in order to be useful to the community.

Reviewer 3 Report

Paper has been improved and it is feasible for publication